# Trends and associations of pulmonary nodule detection rates in China, 2019–2023: A multicenter cross-sectional study based on Real-World Data

Jingxin Li[1☯], Zhouhua Xie[2☯], Yiping Chen[3☯], Guiyun Jin[4☯], Hua Lin[5☯], Qing Xu[6☯], Zhong Meng[7], Lusheng Liang[8], Huiwei Chen[9], Sujuan Guo[10], Xiongwen Li[11], Hao Li[12], Maosheng Liu[13], Youdong Li[14], Yuanzhuang Liao[15], Moyu Ming[16], Shifang Zhou[17], Yang Wu[18], Xikui Huang[19], Wangsheng Deng[20], Yihan Hou[21], Jianfeng Zhang[22], Chaoqian Li[iD][1]*

1 Department of Emergency Medicine/Department of Respiratory Medicine, The First Affiliated Hospital of Guangxi Medical University, Nanning, China, 2 Department of Tuberculosis, The Fourth People's Hospital of Nanning, Nanning, China, 3 Department of General Practice, Guangxi Zhuang Autonomous Region Ethnic Hospital, Nanning, China, 4 Department of Emergency and Interventional Medicine, The First Affiliated Hospital of Hainan Medical University, Haikou, China, 5 Department of Critical Care Medicine, The First Affiliated Hospital of Guangxi University of Chinese Medicine, Nanning, China, 6 Department of General Practice, Guilin People's Hospital, Guilin, China, 7 Department of Radiology, The People's Hospital of Longzhou, Chongzuo, China, 8 Department of Emergency Medicine, Rongxian County People's Hospital, Yulin, China, 9 Department of Respiratory Medicine, Zhuzhou Central Hospital, Zhuzhou, China, 10 Department of Respiratory Medicine, The University of Hong Kong-Shenzhen Hospital, Shenzhen, China, 11 Department of Emergency Medicine, Wuzhou Maternal and Child Health-Care Hospital, Wuzhou, China, 12 Department of Emergency Medicine, The Second Affiliated Hospital of Shandong First Medical University, Tai'an, China, 13 Department of Respiratory Medicine, The First People's Hospital of Zhaoqing, Zhaoqing, China, 14 Department of Emergency Medicine, Xingye County People's Hospital, Yulin, China, 15 Department of Radiology, Qintang District People's Hospital, Guigang, China, 16 Department of Respiratory Medicine, Liuzhou Worker's Hospital, Liuzhou, China, 17 Department of Respiratory Medicine, Changsha Central Hospital, Changsha, China, 18 Department of Respiratory Medicine, The Affiliated Hospital of Southwest Medical University, Luzhou, China, 19 Department of Respiratory Medicine, Beihai People's Hospital, Beihai, China, 20 Department of Emergency Medicine, The People's Hospital of Longhua, Shenzhen, Shenzhen, China, 21 Department of Critical Care Medicine, The First Affiliated Hospital of Xiamen University, Xiamen, China, 22 Department of Emergency Medicine, Wuming Hospital of Guangxi Medical University, Nanning, China

☯ These authors contributed equally to this work.
* lichaoqiangood@163.com

## Abstract

The post-coronavirus disease 2019 (COVID-19) pulmonary sequelae have garnered public concern. We conducted a multicenter cross-sectional study in outpatient and health exam populations from 23 clinical centers (including university-affiliated/provincial general hospitals, municipal general hospitals, county hospitals, and specialized hospitals) in China (2019–2023), to assess temporal trends and potential influencing factors in the detection of CT-diagnosed pulmonary nodules, pleural effusion, pneumonia, and suspected lung tumors, cancer and viral pneumonia, clarifying pandemic impacts on lung health. Dynamic comparisons across key phases including initial outbreak, vaccine rollout, population-wide vaccination, and major adjustment

**Data availability statement:** All relevant data are within the paper and its Supporting Information files.

**Funding:** The author(s) received no specific funding for this work.

**Competing interests:** The authors have declared that no competing interests exist.

of pandemic control policies, were performed. This study analyzed 1,616,750 clinical samples (1,102,605 outpatient, 514,145 health examination; 885,945 males, 730,805 females). Pulmonary nodule detection rose progressively, with surges in 2020−2021 and 2023, plateauing in 2021−2022. Outpatients and males showed steeper increases. University-affiliated/provincial hospitals had sharpest increases vs. municipal and county tiers. Specialized hospitals matched general hospital rates. AI boosted detection rates. CT-suspected lung tumors/cancer remained low and stable, unrelated to nodule trends. These results underscore 2019−2023 pulmonary nodule detection surges linked to SARS-CoV-2 infections and AI adoption. COVID-19 vaccination did not accelerate detection but may have slowed it short-term. Long-term studies on infection, vaccine impacts and pandemic-detected nodules' outcomes are urgently needed.

## Background

The COVID-19 pandemic caused by SARS-CoV-2 infection is still continuing to affect the lives of the public population, although the country has now entered a phase of regularized prevention and control. Based on the complexity and heterogeneity of new-onset disease 30–180 days after SARS-CoV-2 infection, the post-acute sequelae of SARS-CoV-2 infection (PASC) have been identified as four sub-phenotypes in a study led by Dr Rainu Kaushal's research team and published in nature medicine in late 2022 [1,2]. Since then, successive studies have explored the methods of phenotyping PASC [3,4]. However, they have been limited to the symptomatic range, and the mechanisms involved in developing sequelae of SARS-CoV-2 infection need to be further investigated [5–7].

Long COVID was first proposed in 2020 [2]. WHO defines it as persistent symptoms or the appearance of new symptoms that persist for at least 2 months after 3 months of SARS-CoV-2 infection and cannot be explained by any other diagnosis [8]. As the lung is the major target organ of SARS-CoV-2 infection, the infection may cause long COVID-related lung abnormalities [9,10]. Follow-up CT scans one year after COVID-19 diagnosis revealed persistent lung abnormalities in a significant proportion of patients. Studies have reported residual CT abnormalities in 25–47% of patients, with ground-glass opacities and fibrosis-like changes being the most common findings [11–14]. While gradual improvement was observed over time, fibrotic changes showed little improvement between 4–7 months and one year post-infection [13]. These findings highlight the importance of long-term follow-up of COVID-19 patients, and the pulmonary sequelae of SARS-CoV-2 infection remain a widespread community concern, and it has even been suggested that the COVID-19 pandemic may have led to an increase in pulmonary nodules in social groups, but this is not supported by definitive evidence on the basis of the available studies.

In this regard, we first analyzed 2019−2023 chest CT data across three hospital tiers and specialties, revealing yearly increases in pulmonary nodule detection from COVID-19 onset to post-pandemic control. Addressing the lack of national

multicenter studies on pandemic-era trends, we conducted two cross-sectional studies in outpatient and health exam populations to assess pre-/post-pandemic pulmonary nodules and other lung abnormalities rates and influencing factors. This aims to clarify COVID-19's lung health impacts, address public concerns (e.g., infection sequelae), and inform diagnostic and prevention strategies.

## Methods

### Study design

This study is a continuous cross-sectional study, based on the real-world imaging system data of a total of 23 clinical centers, including some university-affiliated/provincial general hospitals, municipal general hospitals, county hospitals, and specialized hospitals (including a cancer hospital, an infectious disease hospital, and a maternity and child healthcare hospital) within China. The number of detections of pulmonary nodules, pleural effusions, and pneumonia diagnosed by chest CT, and the number of detections of suspected lung tumors, lung cancers and viral pneumonia were retrieved in the population of outpatient clinics and medical examinations from 2019 to 2023. The detection rates before and after the cut-off point of each event were compared, and the trends were dynamically analysed.

This study was approved by the Medical Ethical Committee of the First Affiliated Hospital of Guangxi Medical University (2024-K0518, 2024-K0519), all methods were carried out in accordance with the 1964 Helsinki Declaration and its later amendments or comparable ethical standards. The data were accessed for research purposes from 03/01/2025–26/03/2025. The research adheres to anonymity to protect participants' privacy, and no information that could identify participants was collected during data collection, analysis, or reporting stages. Since this study is retrospective and the data used did not involve any personally identifiable information, the need to obtain the informed consent was waived by the First Affiliated Hospital of Guangxi Medical University Medical Ethical Committee.

In accordance with verifiable open-source information, we defined the second half of 2019 as "pre-COVID-19 pandemic", the second half of 2020 as "during COVID-19 pandemic" and "pre-COVID-19 vaccination", the second half of 2021 as "during COVID-19 vaccination", the second half of 2022 as "the universal coverage phase of COVID-19 vaccination" and "pre-regularized prevention and control", and the second half of 2023 as "post-regularized prevention and control", and the time of activating the artificial intelligence(AI) reading was defined according to the specific situation of each hospital. According to the domestic expert consensus, pulmonary nodules are defined as focal, round or oval-shaped solid or subsolid opacities with a maximum diameter ≤ 3 cm on imaging, which are of higher density than lung parenchyma. They can be solitary or multiple, without associated atelectasis, hilar lymphadenopathy, or pleural effusion. Diagnosis of pulmonary nodules, pleural effusion and pneumonia by CT scanning, and suspected diagnosis of lung tumor, lung cancer and viral pneumonia were all made by the Radiology Department in accordance with the standard procedure, whereby the images were first reviewed by an AI system and then by two radiologists of different seniority who specialise in diagnostic CT. The radiologists then independently reviewed the films and signed off on the authentic radiological diagnostic report.

### Statistical analysis

SPSS 26.0 software was used to analyse the data statistically. Count data, including the detection rates, were presented in the form of number of cases/ratios, and differences between groups were compared by chi-square test, Mann-Whitney or Kruskal-Wallis non-parametric test. Logistic regression model was employed to assess the trends of detection rates of pulmonary abnormalities. *P*-value <0.05 was considered statistically significant.

## Results

The study analyzed 1,616,750 samples (1,102,605 outpatients, 514,145 health exams; 885,945 males, 730,805 females) from 23 nationwide centers (2019–2023). Semi-annual pulmonary nodule detection rates rose overall (all *P* for trend

<0.001), with rapid growth in 2020–2021, plateaued in 2021–2022, and surging again in 2023 (Fig 1). While annual rates showed no significant differences between outpatient and health exam groups or genders (all $P > 0.05$), outpatients and males had steeper growth trajectories (Wald $\chi^2$: 29372.929, $P$ for trend < 0.001; Wald $\chi^2$: 16275.107, $P$ for trend < 0.001) (Tables 1 and S4 Table).

Among the 23 hospitals included in our study, 17 have adopted AI-assisted image interpretation. Of these, 3 implemented AI prior to the initial emergence of COVID-19 (pre-2019), 2 after the cessation of the COVID-19 emergency (post-December 2023), and 8 prior to the initiation of COVID-19 vaccination (pre-December 2020). Most showed significant pulmonary nodule detection increases post-AI across populations, while some had non-significant rises or transient declines linked to initial adaptation phases involving workflow adjustments (S1 and S2 Tables). To control AI-related confounding, data from hospitals adopting AI pre-2019, post-2023, and pre-2020 were analyzed. Adjusted trend analyses demonstrated sustained yearly pulmonary nodule detection increases ($P$ for trend <0.001), with moderate pre-pandemic growth (2019), sharp COVID-era surges (2020–21), and stabilization post-2021 amid vaccination rollout. (Fig 2, Tables 2 and S5 Table).

Subgroup analyses were conducted after stratifying 23 participating centers nationwide by institutional tier and specialty classification. The stratified groups comprised 9 university-affiliated/provincial general hospitals, 7 municipal general hospitals, 4 county hospitals, and 3 specialized hospitals (including a cancer hospital, an infectious disease hospital, and a maternity and child healthcare hospital). Specialized hospitals without health exam departments excluded from health

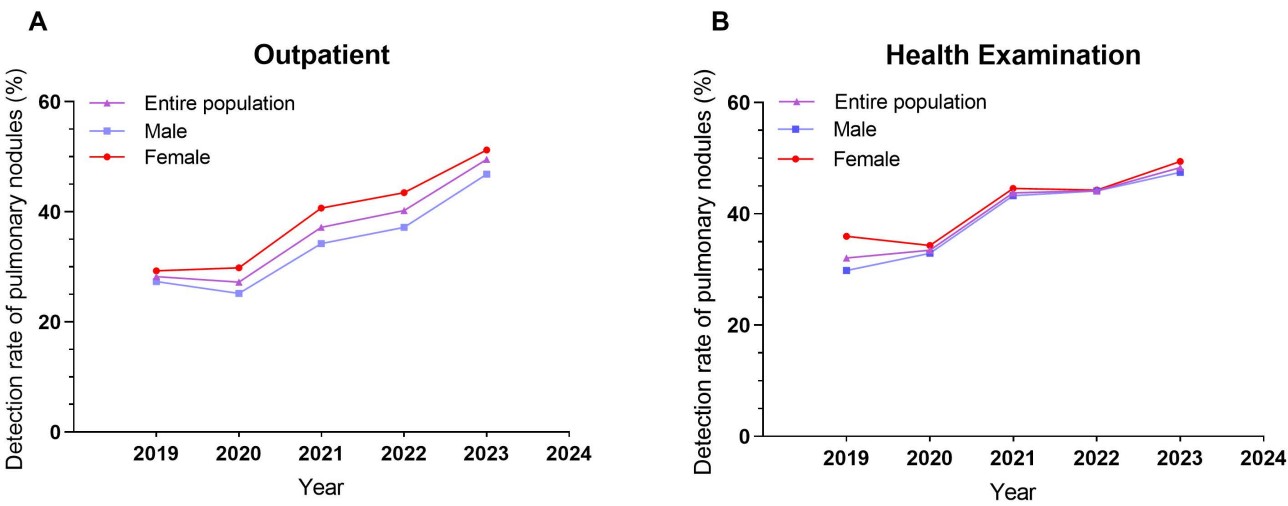

**Fig 1. Trends in Pulmonary Nodule Detection Rates in (A) Outpatients and (B) Health Examination Populations.**

**Table 1. Temporal Trends in Pulmonary Nodule Detection Rates (%) Across Distinct Clinical Populations and Gender Subgroups.**

|  | Years | | | | | | | | |
|---|---|---|---|---|---|---|---|---|---|
|  | **2019** | **2020** | **2021** | **2022** | **2023** | **Wald$\chi^2$** | ***P* for trend** | **U** | ***P*** |
| Outpatient populations | 28.21 | 27.21 | 37.16 | 40.21 | 49.52 | 29372.929 | < 0.001 | 17.000 | 0.421 |
| Male | 27.31 | 25.16 | 34.21 | 37.17 | 46.81 | 16275.107 | < 0.001 |  |  |
| Female | 29.27 | 29.82 | 40.66 | 43.47 | 51.21 | 15265.262 | < 0.001 |  |  |
| Health Examination populations | 32.08 | 33.46 | 43.80 | 44.19 | 48.32 | 7365.155 | < 0.001 | 16.000 | 0.548 |
| Male | 29.83 | 32.93 | 43.27 | 44.14 | 47.47 | 4867.719 | < 0.001 |  |  |
| Female | 35.95 | 34.31 | 44.58 | 44.26 | 49.41 | 2518.971 | < 0.001 |  |  |

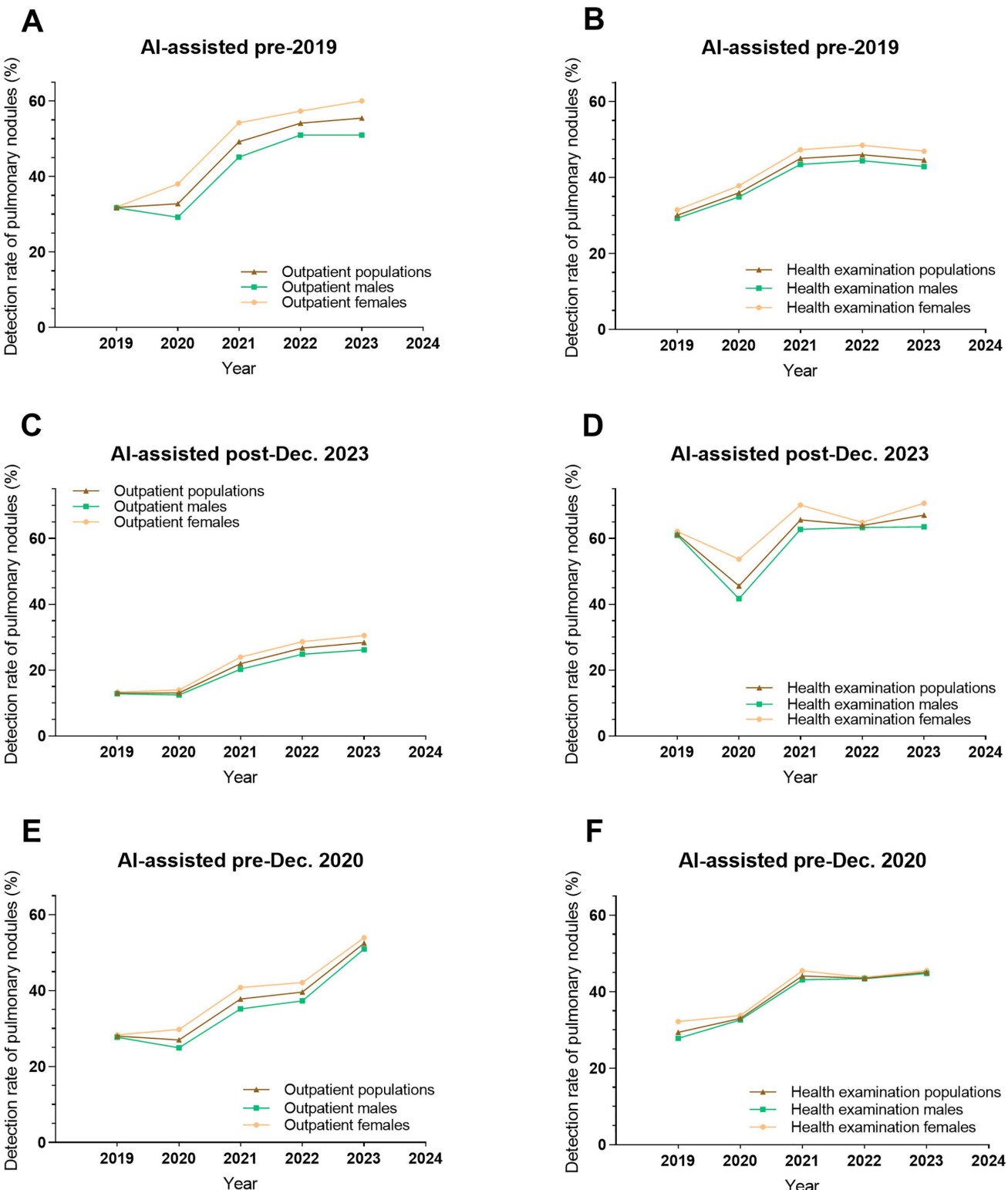

**Fig 2. Trends in pulmonary nodule detection rates among (A) pre-2019 AI-implemented hospitals (outpatients), (B) Pre-2019 AI-implemented hospitals (health examinations), (C) post-Dec. 2023 AI-implemented hospitals (outpatients), (D) post-Dec. 2023 AI-implemented hospitals (health examinations), (E) pre-Dec. 2020 AI-implemented hospitals (outpatients), and (F) pre-Dec. 2020 AI-implemented hospitals (health examinations).**

**Table 2. Temporal Trends in Pulmonary Nodule Detection Rates (%) Across Distinct Clinical Populations and Gender Subgroups after Adjusting for AI-related effects.**

| | Years | | | | | | |
| --- | --- | --- | --- | --- | --- | --- | --- |
| | **2019** | **2020** | **2021** | **2022** | **2023** | **Wald$\chi^2$** | **$P$ for trend** |
| **AI-assisted image interpretation implementation prior to the initial emergence of COVID-19 (pre-2019)** | | | | | | | |
| Outpatient populations | 31.81 | 32.82 | 49.22 | 54.13 | 55.47 | 9696.036 | < 0.001 |
| Male | 31.71 | 29.21 | 45.13 | 51.00 | 50.99 | 4686.051 | < 0.001 |
| Female | 31.93 | 38.04 | 54.26 | 57.31 | 60.05 | 4771.218 | < 0.001 |
| Health Examination populations | 30.10 | 36.00 | 45.05 | 46.02 | 44.61 | 3063.885 | < 0.001 |
| Male | 29.29 | 34.90 | 43.45 | 44.47 | 42.91 | 1726.886 | < 0.001 |
| Female | 31.49 | 37.86 | 47.33 | 48.53 | 46.92 | 1286.656 | < 0.001 |
| **AI-assisted image interpretation implementation post to the cessation of COVID-19 emergency (post-December 2023)** | | | | | | | |
| Outpatient populations | 13.07 | 13.09 | 21.98 | 26.71 | 28.41 | 1551.676 | < 0.001 |
| Male | 12.85 | 12.46 | 20.31 | 24.86 | 26.16 | 717.997 | < 0.001 |
| Female | 13.37 | 13.96 | 23.96 | 28.65 | 30.53 | 781.659 | < 0.001 |
| Health Examination populations | 61.32 | 45.60 | 65.66 | 63.96 | 67.07 | 200.377 | < 0.001 |
| Male | 60.95 | 41.73 | 62.79 | 63.34 | 63.54 | 137.911 | < 0.001 |
| Female | 62.18 | 53.69 | 70.17 | 64.90 | 70.70 | 56.568 | < 0.001 |
| **AI-assisted image interpretation implementation prior to the initiation of COVID-19 vaccination (pre-December 2020)** | | | | | | | |
| Outpatient populations | 28.01 | 27.02 | 37.76 | 39.64 | 52.44 | 23022.052 | < 0.001 |
| Male | 27.72 | 24.97 | 35.18 | 37.28 | 50.94 | 12356.769 | < 0.001 |
| Female | 28.36 | 29.75 | 40.84 | 42.14 | 53.90 | 10289.213 | < 0.001 |
| Health Examination populations | 29.38 | 33.04 | 44.10 | 43.55 | 45.11 | 4497.136 | < 0.001 |
| Male | 27.83 | 32.61 | 43.17 | 43.42 | 44.78 | 3054.310 | < 0.001 |
| Female | 32.19 | 33.79 | 45.46 | 43.76 | 45.51 | 1424.789 | < 0.001 |

exam analysis. All subgroups exhibited significant upward trends in pulmonary nodule detection rates (all *P* for trend <0.001). University-affiliated/provincial general hospitals had sharpest increases across populations and sexes, followed by municipal, then county hospitals. Specialized hospitals matched municipal levels in outpatient gender subgroups (Tables 3 and S6 Table).

Analyses revealed differential patterns across institutional tiers. Among outpatient populations, significant variations in pulmonary nodule detection rates were observed between hospital tiers (male: *P* = 0.033, female: *P* = 0.016), with university-affiliated/provincial general hospitals demonstrating significantly higher detection rates compared to county institutions (male: *P* = 0.033, female: *P* = 0.017). Notably, specialized hospitals exhibited detection rates non-inferior to general hospitals in outpatient populations (all *P* > 0.05).

CT-detected pleural effusion, pneumonia, lung tumors, cancer, and viral pneumonia rates (S3 Table). Specialized (cancer/infectious) hospitals had significant outpatient increases (all *P* for trend<0.05); general hospitals showed no trends (2019−23). Incidental findings in routine health exams (limited specialized data) lacked trends and comparability. COVID-19 temporarily elevated viral pneumonia CT suspicions. Adjusted for AI, lung tumor, cancer rates stabilized, independent of nodule trends (Figs 3 and 4).

## Discussion

The proportion of SARS-CoV-2 infection cases among emergency and outpatient visits in China peaked during the latter half of 2022, followed by a rapid decline and subsequent stabilization beginning in the second half of 2023 [15]. Our statistical analyses further revealed a progressive upward trend in pulmonary nodule detection rates among outpatient and

**Table 3. Temporal Trends in Pulmonary Nodule Detection Rates (%) Across Hospital Tier and Specialty Subgroups.**

| | Years | | | | | | | | |
|---|---|---|---|---|---|---|---|---|---|
| | **2019** | **2020** | **2021** | **2022** | **2023** | **Wald χ²** | **P for trend** | **H** | **P/Adj. P** |
| Outpatient males | | | | | | | | 8.737 | 0.033 [a] |
| University-affiliated/Provincial general hospitals | 31.27 | 28.66 | 36.96 | 38.73 | 50.54 | 9077.117 | < 0.001 | | 0.033 [b] |
| Municipal general hospitals | 26.72 | 21.92 | 34.30 | 37.99 | 42.58 | 4594.631 | < 0.001 | | |
| County hospitals | 16.02 | 18.60 | 19.70 | 22.21 | 23.49 | 107.272 | < 0.001 | | 0.033[b] |
| Specialized hospitals | 17.24 | 21.55 | 27.27 | 33.31 | 40.96 | 2083.69 | < 0.001 | | |
| Outpatient females | | | | | | | | 10.349 | 0.016 [a] |
| University-affiliated/Provincial general hospitals | 32.11 | 32.37 | 42.83 | 46.20 | 53.34 | 9096.226 | < 0.001 | | 0.017 [b] |
| Municipal general hospitals | 28.04 | 27.16 | 41.43 | 40.86 | 48.91 | 3843.258 | < 0.001 | | |
| County hospitals | 13.98 | 18.94 | 19.16 | 23.03 | 22.79 | 119.260 | < 0.001 | | 0.017 [b] |
| Specialized hospitals | 22.02 | 26.13 | 34.63 | 44.34 | 48.19 | 1941.96 | < 0.001 | | |
| Health Examination Males | | | | | | | | 13.018 | 0.005 [a] |
| University-affiliated/provincial general hospitals | 33.00 | 37.21 | 48.66 | 50.41 | 54.68 | 3547.930 | < 0.001 | | |
| Municipal general hospitals | 26.07 | 29.23 | 39.40 | 38.62 | 38.51 | 1563.607 | < 0.001 | | |
| County hospitals | 26.09 | 2.04 | 15.31 | 44·73 | 36.75 | 133.390 | < 0.001 | | |
| Specialized hospitals | 0 | 0 | 0 | 9.09 | 0 | NA | NA | | |
| Health Examination Females | | | | | | | | 16.046 | 0.001 [a] |
| University-affiliated/provincial general hospitals | 39.23 | 37.37 | 48.55 | 47.79 | 53.53 | 1561.044 | < 0.001 | | |
| Municipal general hospitals | 30.50 | 31.37 | 41.63 | 40.59 | 43.47 | 924.588 | < 0.001 | | |
| County hospitals | 17.65 | 6.25 | 13.70 | 30.46 | 30.64 | 32.855 | < 0.001 | | |
| Specialized hospitals | 0 | 0 | 0 | 8.99 | 0 | NA | NA | | |

[a]Kruskal-Wallis test *P*-value.

[b]Bonferroni-corrected Adjusted *P*-value.

health examination populations. Herein, the detection rate of pulmonary nodules is defined as the proportion of outpatients and health examination participants with pulmonary nodules detected via chest CT scans, relative to the total number of individuals who underwent this examination. This trend was marked by a rapid increase between 2020 and 2021, plateaued from 2021 to 2022, and culminated in a renewed surge in 2023.

The observed increase in pulmonary nodule detection rates during the COVID-19 pandemic is multifactorial and may not be solely attributable to SARS-CoV-2 infection [16]. Our study confirmed that AI-assisted image reading significantly increased detection rates. To control AI effects, hospitals were stratified by AI timelines (pre-2019, post-2023, pre-2020). Adjusted pulmonary nodule detection showed gradual rise starting from the early pandemic phase in 2019, sharp surge during the peak of the pandemic (2020–2021), then stabilization post-2021 with vaccination rollout. After China's COVID-19 vaccination rollout (late 2020-early 2021), coverage reached approximately 90% by mid-2022. As vaccination approached 90%, pulmonary nodule detection growth slowed and stabilized, consistent across populations/genders. Findings suggest vaccination did not accelerate detection but may have temporarily slowed it.

Prior to adjusting for AI-related confounding factors, we observed two distinct periods of rapid acceleration in pulmonary nodule detection rates within the trend analysis, the first during the peak of the pandemic (2020–2021) and the second following China's adjustment of pandemic control policies in 2023. The marked rise in detection rates during these periods aligned temporally with surges in population-level infection rates. Notably, after China formally transitioned to regularized prevention and control phase in January 2023, a subset of hospitals adopted AI-assisted image reading systems during this period, which may have contributed to the observed fluctuations in detection rates. Subsequent subgroup analyses controlling for AI implementation confirmed this hypothesis.

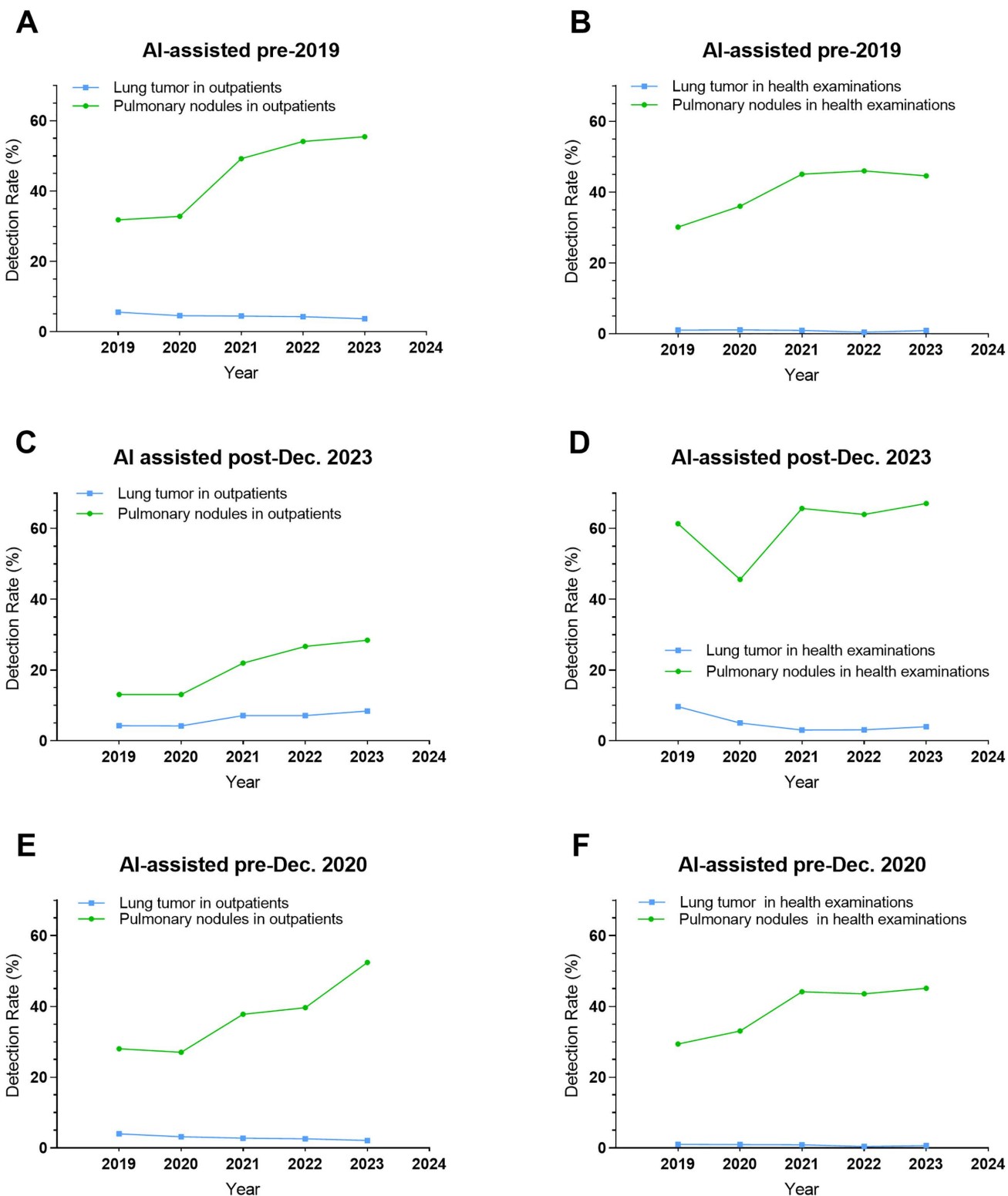

**Fig 3. Comparative trends in pulmonary nodule and lung tumor detection rates across (A) pre-2019 AI-implemented hospitals (outpatients), (B) pre-2019 AI-implemented hospitals (health examinations), (C) post-Dec. 2023 AI-implemented hospitals (outpatients), (D) post-Dec. 2023 AI-implemented hospitals (health examinations), (E) pre-Dec. 2020 AI-implemented hospitals (outpatients), and (F) pre-Dec. 2020 AI-implemented hospitals (health examinations).**

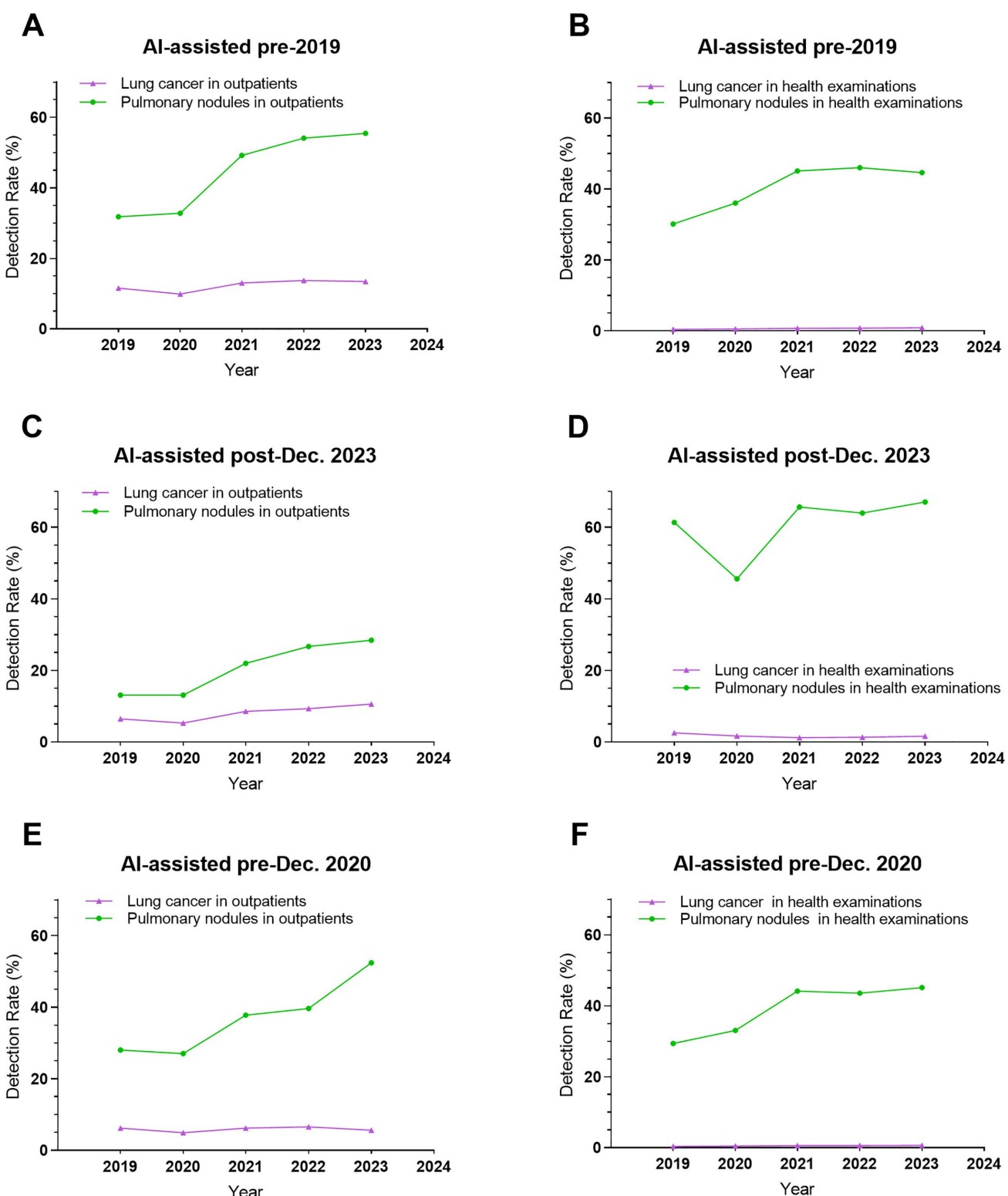

**Fig 4. Comparative trends in pulmonary nodule and lung cancer detection rates across (A) pre-2019 AI-implemented hospitals (outpatients), (B) pre-2019 AI-implemented hospitals (health examinations), (C) post-Dec. 2023 AI-implemented hospitals (outpatients), (D) post-Dec. 2023 AI-implemented hospitals (health examinations), (E) pre-Dec. 2020 AI-implemented hospitals (outpatients), and (F) pre-Dec. 2020 AI-implemented hospitals (health examinations).**

Long-term pulmonary nodule dynamics post-COVID-19 remain unclear. 1-year studies show varied outcomes: some resolve ground-glass and solid nodules from acute infection, others retain persistent lesions, while some develop prolonged abnormalities unrelated to acute-phase pathology [6,13,17–20]. The COVID-19 vaccination-pulmonary nodule link remains debated. Our study suggests vaccination slowed nodule detection short-term (1–2 years), but insufficient data on latency periods preclude linking the 2023 surge to long-term vaccine effects. Further research is needed to clarify causality and temporal relationships.

Our findings suggest that AI-assisted imaging may explain increased pulmonary nodule detection during COVID-19. Pandemic delays in evaluating pulmonary abnormalities including malignant nodules contrasted with expanded diagnostic imaging use. AI addresses this paradox by analyzing vast thoracic imaging data from outpatient and health exam populations, facilitating precise identification of subtle and overlooked nodules, enabling morphological classification and malignancy risk stratification via pattern recognition, and optimizing clinical workflows through lesion prioritization [15]. This synergy reduces diagnostic delays and improves lung surveillance cost-effectiveness. Deep learning algorithms match radiologists in lung cancer risk prediction (AUC 94.4%), cutting false positives by 11% and false negatives by 5% clinically [16]. This technological advancement creates dual opportunities by facilitating incidental identification of early-stage lung cancer through enhanced diagnostic sensitivity, and optimizing management pathways for benign nodules by developing models that accurately predict malignant potential, thereby reducing diagnostic delays, minimizing false reassurance risks, and alleviating patient anxiety [21]. Furthermore, AI has been extensively integrated into COVID-19-related solutions spanning epidemic forecasting, viral transmission tracking, diagnostic and therapeutic interventions, as well as vaccine discovery and pharmaceutical research [22,23]. However, despite its widespread adoption in medical imaging interpretation, AI systems remain susceptible to diagnostic inaccuracies under specific circumstances due to multifaceted limitations-a reality necessitating critical perspective when interpreting algorithmic outputs [24].

COVID-19 masks were vital for prevention but sparked debate on pulmonary nodule links. No direct causal link established. Studies confirm mask safety [25,26]. While some suggest prolonged use may alter respiratory microenvironments (humidity and airflow), but no data proves this causes nodules [27].

Pulmonary nodule detection rates vary by hospital tier and specialty. University-affiliated/provincial general hospitals show higher baseline rates and faster growth than county hospitals, likely due to superior diagnostics, complex case management (critical and referred patients), and post-COVID AI adoption (absent in county hospitals). Specialized hospitals, though limited in routine screenings, matched general hospitals' outpatient detection trends during COVID-19. Their focus on detailed nodule characterization via advanced diagnostics may boost sensitivity, but direct rate comparisons are confounded by high-risk and symptomatic patient cohorts.

Studies found non-COVID respiratory viruses declined during the pandemic, likely due to COVID-related non-pharmaceutical interventions (NPIs) suppressing pathogen circulation [28,29]. Outpatient clinics had brief rises in CT-suspected viral pneumonia during the pandemic, lacking confirmatory etiological testing. Post-pandemic studies are needed, as SARS-CoV-2's enhanced lung invasiveness and replication efficiency may underlie this discrepancy [30,31], we hypothesize SARS-CoV-2 may alter co-occurring respiratory viruses' pathogenicity, affecting pulmonary tropism or modifying radiological lung injury patterns.

During the COVID-19 pandemic, the detection rates of CT-suspected lung tumors and lung cancer remained low and stable, with no apparent correlation observed compared to the rising trend in pulmonary nodule detection rates. In subsequent phases, we will conduct long-term follow-up studies combined with radiological and pathological evaluations of populations with detected pulmonary nodules to elucidate the ultimate clinical outcomes of nodules identified during the pandemic period.

Our study has certain limitations. First, the detection rate of pulmonary nodules in this paper reflects the proportion of nodules detected via imaging in a specific population, and its results may depend on AI detection technology, detection frequency, and population willingness to undergo testing. For the reasons mentioned above, an increase in

detection rate does not necessarily indicate a rise in disease prevalence. Therefore, the subsequent cohort studies are crucial. In addition, considering that AI may identify nodules that were previously undetected, although we attempted to exclude these confounding factors through stratified analysis of hospital grades and AI adoption status, the "detection rate" might be closer to the true "prevalence" if the authenticity of nodules could be verified based on clear pathological diagnosis or long-term follow-up. In the future, we aim to adopt combined imaging-pathological diagnosis and extend the follow-up period to avoid conclusion bias caused by conceptual confusion. Secondly, although this study speculates that the peak of the pandemic may be related to the increased rate of lung nodule detection, and that vaccines may have slowed this rate, there are inherent limitations in tracking individual-level infection details and vaccination status. This limitation prevents us from establishing a direct causal relationship between individual lung nodule detection and personal infection or vaccination history. In follow-up studies, we plan to conduct cohort studies and case-control studies to perform a detailed assessment from baseline levels, in order to supplement and validate the population-level trends observed in this study.

Our study also identified several unresolved questions. First, unclear timing of infection-induced nodule formation, such as latency and persistence. Second, vaccination's long-term nodule effects unvalidated, potential post-vaccination intervals unknown. Furthermore, pandemic CT-suspected viral pneumonia spikes lacked etiological confirmation, requiring analysis of whether SARS-CoV-2 alters co-pathogen virulence and imaging injury patterns. Finally, addressing public concerns, longitudinal follow-up with imaging and pathology will clarify clinical outcomes of pandemic-detected nodules.

## Conclusions

From 2019−2023, pulmonary nodule detection rates rose progressively in outpatient and health exam populations, with two surges: during COVID-19 peaks (2020−2021) and post-China's 2023 policy transition. While linked to infection dynamics, AI adoption's confounding role was noted. Post-AI adjustment, growth slowed and stabilized post-2021 amid vaccination expansion (90% coverage by mid-2022), suggesting vaccines temporarily decelerated detection. Mask-wearing may explain the early dip (2019−2020) but not later fluctuations. Higher-tier hospitals surpassed primary care in rates, while specialized hospitals matched general ones during COVID-19 but required rigorous nodule analysis. Lung tumor and cancer detection remained stable, unrelated to nodule trends.

These results indicate that there was a surge in the detection of pulmonary nodules between 2019 and 2023, which may be related to SARS-CoV-2 infection and the application of AI. At the same time, vaccination may have slowed down the detection. However, the inherent limitations of this study also suggest that, although public concern about pulmonary nodules is understandable, an increase in detection does not necessarily mean an increase in prevalence, and changes in detection rates may be influenced by multiple factors, which requires further validation of their correlations.

## Supporting information

**S1 Table. Impact of AI-Assisted Image Interpretation on Pulmonary Nodule Detection Rates (%) in Outpatient Populations.**
(DOCX)

**S2 Table. Impact of AI-Assisted Image Interpretation on Pulmonary Nodule Detection Rates (%) in Health Examination Populations.**
(DOCX)

**S3 Table. Temporal Trends in CT-Diagnosed Pleural Effusion, Pneumonia, Suspected Lung Tumors, Lung Cancer, and Viral Pneumonia Detection Rates Across Distinct Subgroups.**
(DOCX)

**S4 Table. Temporal Trends in Pulmonary Nodule Detection Rates (%) Across Distinct Clinical Populations and Gender Subgroups.** (Including Number of Cases/ Total Samples).
(DOCX)

**S5 Table. Temporal Trends in Pulmonary Nodule Detection Rates (%) Across Distinct Clinical Populations and Gender Subgroups after Adjusting for AI-related effects.** (Including Number of Cases/ Total Samples).
(DOCX)

**S6 Table. Temporal Trends in Pulmonary Nodule Detection Rates (%) Across Hospital Tier and Specialty Subgroups.** (Including Number of Cases/ Total Samples).
(DOCX)

## Author contributions

**Conceptualization:** Chaoqian Li.

**Data curation:** Chaoqian Li.

**Formal analysis:** Jingxin Li.

**Investigation:** Jingxin Li, Zhouhua Xie, Yiping Chen, Guiyun Jin, Hua Lin, Qing Xu.

**Methodology:** Jingxin Li, Zhouhua Xie, Yiping Chen, Guiyun Jin, Hua Lin, Qing Xu.

**Project administration:** Zhouhua Xie, Yiping Chen, Zhong Meng, Lusheng Liang, Huiwei Chen, Sujuan Guo, Xiongwen Li, Hao Li, Maosheng Liu, Youdong Li, Yuanzhuang Liao, Moyu Ming, Shifang Zhou, Yang Wu, Xikui Huang, Wangsheng Deng, Yihan Hou, Jianfeng Zhang.

**Resources:** Chaoqian Li, Jingxin Li, Yiping Chen, Guiyun Jin, Hua Lin, Qing Xu, Zhong Meng, Lusheng Liang, Huiwei Chen, Sujuan Guo, Xiongwen Li, Hao Li, Maosheng Liu, Youdong Li, Yuanzhuang Liao, Moyu Ming, Shifang Zhou, Yang Wu, Xikui Huang, Wangsheng Deng, Yihan Hou, Jianfeng Zhang.

**Software:** Chaoqian Li, Jingxin Li.

**Supervision:** Chaoqian Li.

**Validation:** Jingxin Li, Zhouhua Xie.

**Visualization:** Jingxin Li.

**Writing – original draft:** Jingxin Li.

**Writing – review & editing:** Chaoqian Li.

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
