## [Decision Letter · Decision Letter 0]

9 Jan 2026

Dear Dr. Li,

Thank you for submitting your manuscript to PLOS ONE. After careful consideration, we feel that it has merit but does not fully meet PLOS ONE’s publication criteria as it currently stands. Therefore, we invite you to submit a revised version of the manuscript that addresses the points raised during the review process.

The reviewers have recommended publication, but also suggest significant revisions to your manuscript.  Therefore, I invite you to respond to the reviewers' comments and revise your manuscript.

We look forward to receiving your revised manuscript.

Kind regards,

Fumihiro Yamaguchi

Academic Editor

PLOS One

Journal Requirements:

“This work was supported by the National Science & Technology Fundamental Resources Investigation Program of China (2018FY100603) and Guangxi Science and Technology Major Program (Guikegong 14124003-7).”

“This work was supported by the National Science & Technology Fundamental Resources Investigation Program of China (2018FY100603) and Guangxi Science and Technology Major Program (Guikegong 14124003-7).”

Reviewers' comments:

Reviewer's Responses to Questions

**Comments to the Author**

1. Is the manuscript technically sound, and do the data support the conclusions?

Reviewer #1: Partly

Reviewer #2: Yes

2. Has the statistical analysis been performed appropriately and rigorously?

Reviewer #1: Yes

Reviewer #2: Yes

3. Have the authors made all data underlying the findings in their manuscript fully available?

Reviewer #1: Yes

Reviewer #2: Yes

4. Is the manuscript presented in an intelligible fashion and written in standard English?

Reviewer #1: Yes

Reviewer #2: Yes

Reviewer #1: I acknowledge the ambition and scale of this multicenter cross-sectional study. The topic of post-COVID-19 pulmonary sequelae and the impact of the pandemic on lung health are undoubtedly crucial. However, for this manuscript to achieve its envisioned position in the field, several fundamental epidemiological and methodological deficiencies need to be addressed. My critique focuses on areas where the study's design and interpretation fall short of robust epidemiological standards, particularly when drawing conclusions about "determinants" and causal links.

Major concerns:

1. Limitations of Cross-Sectional Design for Causal Inference:

a. I am particularly concerned about the use of a strong word, "Determinants" in this study title! Of note, keenly considering "Determinants" vs. "Associations", here explicitly suggesting a causal relationship. However, a multicenter cross-sectional study, even with temporal trend analysis, is inherently limited in establishing causality. It can identify associations and trends, but cannot definitively prove that SARS-CoV-2 infection or vaccination caused or prevented specific changes in nodule incidence. The observed "surges" are co-temporal with pandemic events and AI adoption, but co-occurrence does not equal causation. For a "great" paper in epidemiology, the language must reflect the true strength of the evidence.

b. Also, in terms of “Reverse Causality/Bias”, could changes in health-seeking behavior, increased awareness due to the pandemic, or altered screening guidelines contribute to increased detection rates rather than an actual increase in nodule prevalence or incidence? This is a significant confounding factor that cross-sectional data struggles to disentangle.

2. Lack of Individual-Level Data on Key Exposure and Confounding Variables:

a. The study correlates nodule detection trends with population-level "surges in population-level infection rates" and "pandemic peaks." However, there's no individual-level data on who was infected, when, or with what severity. Without this, linking individual nodule detection to individual infection history is impossible. Many observed nodules could be incidental, pre-existing, or related to other factors putting the conclusion at the risk of misleading the public and particularly the policy makers.

b. Similarly, the conclusion that "COVID-19 vaccination did not accelerate detection but may have slowed it short-term" is based on population-level vaccination rollout periods. There's no data to show the vaccination status of individuals with detected nodules. This is a critical gap for making any claim about vaccine impact. Observational data at a population level is highly susceptible to ecological fallacy and unmeasured confounders.

c. There are several other confounders noted in the paper. For instance, for pulmonary nodules, factors like smoking history, age, occupation, environmental exposures, and pre-existing lung conditions are paramount. The study mentions "health exam populations" which might differ significantly from "outpatient populations" in these baseline characteristics. Without adjusting for these at the individual level, it's difficult to attribute observed trends solely to COVID-19-related factors or AI.

3. "Detection Rate" vs. "Incidence/Prevalence":

a. I noticed that the manuscript consistently reported “pulmonary nodule detection rates." An increase in detection can be driven by improved technology (AI), increased screening volume, or greater scrutiny by radiologists, independent of a true increase in nodule incidence. The paper acknowledges AI's role as a confounder and attempts to adjust for it, which is good, but the overall increase in CT scans during the pandemic (due to respiratory symptoms) could also inflate detection. This distinction needs to be central to the interpretation, especially for epidemiological conclusions.

4. Definition and Standardization of "Pulmonary Nodule":

a. The manuscript states nodules were "CT-diagnosed" and reviewed by AI and radiologists. However, specific criteria for what constitutes a "pulmonary nodule" (size, morphology, consistency) and the follow-up protocols might vary across 23 diverse centers (university-affiliated, municipal, county, specialized hospitals). This potential lack of standardization could introduce heterogeneity and bias into the aggregated "detection rates." More information to accurately portray standardized term by definitive criteria is required.

5. Generalizability and Representativeness of the Study Population:

a. The study analyzes data from "outpatient and health exam populations" from 23 centers. While substantial, these populations may not be representative of the general Chinese population. Outpatients are by definition seeking medical care, often due to symptoms, while health exam participants may be self-selected or screened for specific purposes. This limits the generalizability of the "detection rates" to the broader population and could risk erroneous outcome and mislead the public.

6. Mechanistic Speculation Without Direct Evidence:

a. The discussion touches upon mechanisms like pulmonary fibrosis and microvascular damage as pathophysiological bases for Long COVID. While relevant to the broader field, the current study does not provide direct evidence linking these mechanisms to the detected pulmonary nodules in their study population. The findings are primarily statistical trends of detection, not pathological characterizations. The authors need to address this against overstretching the fact beyond their data.

7. Regarding the Timing of Ethical Data Access:

a. Lines 130-131: The methods state: "The data were accessed for research purposes from 03/01/2025 to 26/03/2025." Given the server time is 2025-12-03, this indicates the data access period has not yet occurred. While possibly a typo (e.g., 2024), this raises a flag about the practical feasibility of the study timeframe and reflects a potential lack of precision in documentation. If it's indeed data accessed in the future, it undermines the credibility of the research as presented.

Minor Concerns:

1. Redundancy of "P for trend <0.001" was noted: While important, simply stating "<0.001" for every P-value makes it harder to discern relative significance or subtle differences if they exist. More precise P-values (e.g., 1.2e-5, 0.0003) would be more informative.

2. The table presents (number of cases / total samples). While clear, the overall sample sizes for each year/subgroup are very large. This level of detail is good, but ensuring readability with large numbers is key.

Reviewer #2: Thank you for the opportunity to review this interesting manuscript on the topic “Trends and Determinants of Pulmonary Nodule detection Rates in China, 2019-2023: A Multicenter Cross-Sectional Study Based on Real-World Data.”

This research evaluates the rate of detection of lung nodules from CT scans conducted in China between 2019 and 2023, based on data gathered from over 1.6 million scans from 23 different hospitals. This issue is significant because lung nodules remain a significant health issue, particularly in the wake of COVID-19.

The research is well designed with a massive dataset. Nevertheless, some findings could have been clarified in a more understandable manner.

1. Introduction is fine with the background being explained fully; however, it does not clearly define the main study purpose. A statement with respect to the purpose would give an instant view on why this study was conducted.

2. In the Methods, the data origin and size are described in a clear manner. The image-reading workflow is also explained. However, this section is quite lengthy and incorporates a lot of professional jargon that can be hard to understand particularly for readers who are not specialists in radiology or AI-assisted imaging.

3. The Results show a clear overall improvement in detection rates. However, it can be difficult for the reader to gauge what they should give the highest importance to. Adding sentences to summarize "What has changed" and "Why it matters" would greatly improve the clarity.

4. The manuscript states that AI has improved the rate of detection considerably. This fact is crucial; however, it may also be misinterpreted to mean a higher prevalence of the disease, as opposed to improved diagnosis. Explicitly declare that AI may identify nodules which were not seen before and "Increased detection does not necessarily mean increased prevalence."

5. Certain limitations are also highlighted, albeit disseminated over the Discussion section. It could help in adding a dedicated paragraph on limitations.

**Do you want your identity to be public for this peer review?** For information about this choice, including consent withdrawal, please see our Privacy Policy

Reviewer #1: **Yes:** Kamoru A. Adedokun

Reviewer #2: No

---

## [Author Response · Author response to Decision Letter 1]

15 Jan 2026

Reviewer #1:

Comment 1:

Limitations of Cross-Sectional Design for Causal Inference:

a. I am particularly concerned about the use of a strong word, "Determinants" in this study title! Of note, keenly considering "Determinants" vs. "Associations", here explicitly suggesting a causal relationship. However, a multicenter cross-sectional study, even with temporal trend analysis, is inherently limited in establishing causality. It can identify associations and trends, but cannot definitively prove that SARS-CoV-2 infection or vaccination caused or prevented specific changes in nodule incidence. The observed "surges" are co-temporal with pandemic events and AI adoption, but co-occurrence does not equal causation. For a "great" paper in epidemiology, the language must reflect the true strength of the evidence.

Response 1a: First of all, we believe it is very necessary to carefully consider your suggestion regarding the use of the term 'determinants' in the title. Our study is a cross-sectional study combined with time trend analysis. By describing the changes in the detection rate of lung nodules and matching these trends with significant events such as SARS-CoV-2 infection, vaccination, and AI implementation, we aim to infer that these events may be associated with changes in lung nodule detection trends. At the same time, we also hope that this cross-sectional study can preliminarily explore factors that may influence the detection of lung nodules, thereby providing a foundation for subsequently clarifying causal relationships through methods such as case-control studies or animal experiments. In summary, we have decided to change 'determinants' to 'associations' in the title to reflect the rigor expected in epidemiological research.

b. Also, in terms of “Reverse Causality/Bias”, could changes in health-seeking behavior, increased awareness due to the pandemic, or altered screening guidelines contribute to increased detection rates rather than an actual increase in nodule prevalence or incidence? This is a significant confounding factor that cross-sectional data struggles to disentangle.

Response 1b: We fully agree with your viewpoint that “detection rate” cannot be equated entirely with “prevalence”. The original intention of our study was to objectively present the actual detection of nodules in CT scans during the study period using real-world data; therefore, we chose the indicator “detection rate.” We also recognize that due to reasons such as population selection bias, there may be misjudgments regarding disease prevalence trends. To avoid conceptual confusion, we have added a detailed explanation of the concept of “detection rate” in the article, and in the limitations section of the study, we pointed out that “detection rate cannot be completely equated with prevalence” and provided corresponding solutions. The revisions are as follows:

Lines 251-254, provide a detailed explanation of the concept of “detection rate” in this study.

Herein, the detection rate of pulmonary nodules is defined as the proportion of outpatients and health examination participants with pulmonary nodules detected via chest CT scans, relative to the total number of individuals who underwent this examination.

Lines 335-346, clarify that “detection rate” in this study cannot be completely equated with “prevalence,” and propose solutions to make the “detection rate” more similar to “prevalence.”

Our study has certain limitations. First, the detection rate of pulmonary nodules in this paper reflects the proportion of nodules detected via imaging in a specific population, and its results may depend on AI detection technology, detection frequency, and population willingness to undergo testing. For the reasons mentioned above, an increase in detection rate does not necessarily indicate a rise in disease prevalence. Therefore, the subsequent cohort studies are crucial. In addition, considering that AI may identify nodules that were previously undetected, although we attempted to exclude these confounding factors through stratified analysis of hospital grades and AI adoption status, the "detection rate" might be closer to the true "prevalence" if the authenticity of nodules could be verified based on clear pathological diagnosis or long-term follow-up. In the future, we aim to adopt combined imaging-pathological diagnosis and extend the follow-up period to avoid conclusion bias caused by conceptual confusion.

2. Lack of Individual-Level Data on Key Exposure and Confounding Variables:

a. The study correlates nodule detection trends with population-level "surges in population-level infection rates" and "pandemic peaks." However, there's no individual-level data on who was infected, when, or with what severity. Without this, linking individual nodule detection to individual infection history is impossible. Many observed nodules could be incidental, pre-existing, or related to other factors putting the conclusion at the risk of misleading the public and particularly the policy makers.

Response 2a: First of all, we sincerely thank you for raising the concern about the lack of individual-level infection data. This comment accurately points out the key limitation in verifying the association between lung nodule detection and SARS-CoV-2 infection, and we fully agree with it. Our study was designed as a population-level macro data study, which inherently limits the tracking of individual-level infection details. This limitation indeed prevents us from establishing a direct causal link between the detection of lung nodules and personal infection history. Therefore, we have added this point to the limitations section of our study and proposed potential solutions to address this gap. The revisions are on Lines 346-354.

Secondly, although this study speculates that the peak of the pandemic may be related to the increased rate of lung nodule detection, and that vaccines may have slowed this rate, there are inherent limitations in tracking individual-level infection details and vaccination status. This limitation prevents us from establishing a direct causal relationship between individual lung nodule detection and personal infection or vaccination history. In follow-up studies, we plan to conduct cohort studies and case-control studies to perform a detailed assessment from baseline levels, in order to supplement and validate the population-level trends observed in this study.

b. Similarly, the conclusion that "COVID-19 vaccination did not accelerate detection but may have slowed it short-term" is based on population-level vaccination rollout periods. There's no data to show the vaccination status of individuals with detected nodules. This is a critical gap for making any claim about vaccine impact. Observational data at a population level is highly susceptible to ecological fallacy and unmeasured confounders.

Response 2b: We strongly agree with your suggestion. Similar to Response 2a, we have added this content in the limitations section and plan to address this limitation in future research through a case-control study. Our revisions are on Lines 346-354.

c. There are several other confounders noted in the paper. For instance, for pulmonary nodules, factors like smoking history, age, occupation, environmental exposures, and pre-existing lung conditions are paramount. The study mentions "health exam populations" which might differ significantly from "outpatient populations" in these baseline characteristics. Without adjusting for these at the individual level, it's difficult to attribute observed trends solely to COVID-19-related factors or AI.

Response 2c: We agree with your suggestion that these factors are crucial for interpreting trends in the detection of pulmonary nodules, as they may independently influence the risk of nodule occurrence and healthcare-seeking behavior. Given that this study involves a large sample size and the data were sourced from the imaging systems of various centers, our data collection phase was somewhat limited in obtaining the detailed baseline information mentioned above. We hope that through subsequent cohort and case-control studies, we can perform a comprehensive integration of the baseline characteristics of the population to address the limitations of the current study.

3. "Detection Rate" vs. "Incidence/Prevalence":

a. I noticed that the manuscript consistently reported “pulmonary nodule detection rates." An increase in detection can be driven by improved technology (AI), increased screening volume, or greater scrutiny by radiologists, independent of a true increase in nodule incidence. The paper acknowledges AI's role as a confounder and attempts to adjust for it, which is good, but the overall increase in CT scans during the pandemic (due to respiratory symptoms) could also inflate detection. This distinction needs to be central to the interpretation, especially for epidemiological conclusions.

Response 3: As the reviewer pointed out, changes in the detection rate of lung nodules in this study need to be interpreted with caution due to non-disease-related factors, such as an increase in outpatient visits for respiratory symptoms during the pandemic and heightened public awareness of health screening, both of which led to a significant rise in CT scans. The increased opportunities for examination may have raised the likelihood of detecting nodules, rather than directly reflecting an actual change in the risk of lung nodule occurrence. In this study, the term “detection rate” refers to the proportion of individuals undergoing CT scans within a specific time period who are diagnosed with lung nodules. This outcome directly depends on the coverage of examinations, diagnostic technologies such as AI-assisted diagnosis, and physician diagnostic standards, and can only reflect nodule detection within the context of examination behavior. In contrast, “incidence” requires baseline data from a representative population and is calculated as the proportion of new cases over a certain period, reflecting the risk of new disease occurrence. “Prevalence”, on the other hand, refers to the proportion of existing cases (including both new and old cases) in a population at a specific point in time. Both indicators require strict control over population representativeness and elimination of examination opportunity bias to accurately reflect the true epidemiological status of the disease. This study chose “detection rate” as the core metric precisely due to careful consideration of these confounding factors. On one hand, the population undergoing CT scans during the study period was not a randomly selected community population but included a large number of patients visiting due to respiratory symptoms and individuals undergoing voluntary health checkups, whose baseline disease risk differs significantly from the general population, lacking the population representativeness required to calculate “incidence” or “prevalence”. On the other hand, due to the absence of long-term follow-up data for the same population, it was impossible to distinguish between “newly detected nodules” and “previously existing but undetected nodules”, which further limits the applicability of the “incidence” metric. Therefore, using the “detection rate” avoids misjudgment of disease trends caused by selection bias and lack of follow-up data, and more objectively reflects the actual detection of nodules in CT exams during the study period. Accordingly, to avoid conceptual confusion, we provided a detailed explanation in the discussion section of the article. The revisions are as follows:

Lines 251-254, provide a detailed explanation of the concept of “detection rate” in this study.

Herein, the detection rate of pulmonary nodules is defined as the proportion of outpatients and health examination participants with pulmonary nodules detected via chest CT scans, relative to the total number of individuals who underwent this examination.

Lines 335-346, clarify that “detection rate” in this study cannot be completely equated with “prevalence,” and propose solutions to make the “detection rate” more similar to “prevalence.”

Our study has certain limitations. First, the detection rate of pulmonary nodules in this paper reflects the proportion of nodules detected via imaging in a specific population, and its results may depend on AI detection technology, detection frequency, and population willingness to undergo testing. For the reasons mentioned above, an increase in detection rate does not necessarily indicate a rise in disease prevalence. Therefore, the subsequent cohort studies are crucial. In addition, considering that AI may identify nodules that were previously undetected, although we attempted to exclude these confounding factors through stratified analysis of hospital grades and AI adoption status, the "detection rate" might be closer to the true "prevalence" if the authenticity of nodules could be verified based on clear pathological diagnosis or long-term follow-up. In the future, we aim to adopt combined imaging-pathological diagnosis and extend the follow-up period to avoid conclusion bias caused by conceptual confusion.

4. Definition and Standardization of "Pulmonary Nodule":

a. The manuscript states nodules were "CT-diagnosed" and reviewed by AI and radiologists. However, specific criteria for what constitutes a "pulmonary nodule" (size, morphology, consistency) and the follow-up protocols might vary across 23 diverse centers (university-affiliated, municipal, county, specialized hospitals). This potential lack of standardization could introduce heterogeneity and bias into the aggregated "detection rates." More information to accurately portray standardized term by definitive criteria is required.

Response 4: In this study, the definition of "pulmonary nodule" is based on the consensus of domestic experts and is uniformly defined as a focal, roundish shadow with a maximum diameter of ≤3 cm, showing increased density relative to lung parenchyma, which can be solid or subsolid. It may be solitary or multiple and is not accompanied by atelectasis, hilar lymphadenopathy, or pleural effusion. We added an explanation of this standard term in the Methods section, Lines 143-147.

According to the domestic expert consensus, pulmonary nodules are defined as focal, round or oval-shaped solid or subsolid opacities with a maximum diameter ≤ 3 cm on imaging, which are of higher density than lung parenchyma. They can be solitary or multiple, without associated atelectasis, hilar lymphadenopathy, or pleural effusion.

5. Generalizability and Representativeness of the Study Population:

a. The study analyzes data from "outpatient and health exam populations" from 23 centers. While substantial, these populations may not be representative of the general Chinese population. Outpatients are by definition seeking medical care, often due to symptoms, while health exam participants may be self-selected or screened for specific purposes. This limits the generalizability of the "detection rates" to the broader population and could risk erroneous outcome and mislead the public.

Response 5: Thank you for your suggestion. We selected outpatient and health check-up populations from 23 centers because these two groups are the main populations in which clinical lung nodules are detected. However, selection bias does indeed exist, and these two groups cannot represent the characteristics of the general population, making our study results applicable only to these two populations and inherently limited. Therefore, similar to our response to comment 3, we emphasized in the discussion section that the “detection rate” in this study is not fully equivalent to the “prevalence/incidence” in the general population, to avoid any misunderstanding.

6. Mechanistic Speculation Without Direct Evidence:

a. The discussion touches upon mechanisms like pulmonary fibrosis and microvascular damage as pathophysiological bases for Long COVID. While relevant to the broader field, the current study does not provide direct evidence linking these mechanisms to the detected pulmonary nodules in their study population. The findings are primarily statistical trends of detection, not pathological characterization

---

## [Decision Letter · Decision Letter 1]

4 Feb 2026

Trends and Associations of Pulmonary Nodule Detection Rates in China, 2019-2023: A Multicenter Cross-Sectional Study Based on Real-World Data

PONE-D-25-59979R1

Dear Dr. Li,

We’re pleased to inform you that your manuscript has been judged scientifically suitable for publication and will be formally accepted for publication once it meets all outstanding technical requirements.

Kind regards,

Fumihiro Yamaguchi

Academic Editor

PLOS One

Additional Editor Comments (optional):

Reviewers' comments:

Reviewer's Responses to Questions

**Comments to the Author**

Reviewer #1: All comments have been addressed

Reviewer #2: All comments have been addressed

2. Is the manuscript technically sound, and do the data support the conclusions?

Reviewer #1: Yes

Reviewer #2: Yes

3. Has the statistical analysis been performed appropriately and rigorously?

Reviewer #1: Yes

Reviewer #2: Yes

4. Have the authors made all data underlying the findings in their manuscript fully available?

Reviewer #1: Yes

Reviewer #2: Yes

5. Is the manuscript presented in an intelligible fashion and written in standard English?

Reviewer #1: Yes

Reviewer #2: Yes

Reviewer #1: The response satisfies my critiques having provided appropriate answers where necessary. Therefore, I have no other comments. Thank you.

Reviewer #2: Accept it as it is; all comments were addressed.

Accept it as it is; all comments were addressed.

Accept it as it is; all comments were addressed.

**Do you want your identity to be public for this peer review?** For information about this choice, including consent withdrawal, please see our Privacy Policy

Reviewer #1: **Yes:** Kamoru Adedokun, BMLS, MSC., AMLSCN,

Reviewer #2: No

---

## [Editor Report · Acceptance letter]

PONE-D-25-59979R1

PLOS One

Dear Dr. Li,

I'm pleased to inform you that your manuscript has been deemed suitable for publication in PLOS One. Congratulations! Your manuscript is now being handed over to our production team.

Kind regards,

on behalf of

Dr. Fumihiro Yamaguchi

Academic Editor

PLOS One